# Design of Recyclable Carboxylic Metal-Organic Framework/Chitosan Aerogels for Oil Bleaching

**DOI:** 10.3390/foods12224151

**Published:** 2023-11-16

**Authors:** Xiang-Ze Jia, Qing-Bo Yao, Bin Zhang, Chin-Ping Tan, Xin-An Zeng, Yan-Yan Huang, Qiang Huang

**Affiliations:** 1Guangdong Provincial Key Laboratory of Intelligent Food Manufacturing, College of Food Science and Engineering, Foshan University, Foshan 528225, China; yqb123999@163.com (Q.-B.Y.); xazeng@scut.edu.cn (X.-A.Z.); 2Guangdong Province Key Laboratory for Green Processing of Natural Products and Product Safety, School of Food Science and Engineering, South China University of Technology, Guangzhou 510640, China; xiangze1992@163.com (X.-Z.J.); zhangb@scut.edu.cn (B.Z.); 3SCUT-Zhuhai Institute of Modern Industrial Innovation, Zhuhai 519175, China; 4Overseas Expertise Introduction Center for Discipline Innovation of Food Nutrition and Human Health (111 Center), Guangzhou 510640, China; tancp@upm.edu.my; 5Department of Food Technology, Faculty of Food Science and Technology, Universiti Putra Malaysia, Serdang 43400, Malaysia

**Keywords:** metal organic framework, chitosan, aerogel, oil bleaching, reusability

## Abstract

Novel hierarchical metal-organic framework/chitosan aerogel composites were developed for oil bleaching. UiO-66-COOH-type metal organic frameworks (Zr-MOFs) were synthesized and integrated onto a chitosan matrix with different contents and named MOF-aerogel-1 and MOF-aerogel-2. Due to the compatibility of chitosan, the carboxylic zirconium MOF-aerogels not only maintained the inherent chemical accessibility of UiO-66-COOH, but the unique crystallization and structural characteristics of these MOF nanoparticles were also preserved. Through 3-dimensional reconstructed images, aggregation of the UiO-66-COOH particles was observed in MOF-aerogel-1, while the MOF was homogeneously distributed on the surface of the chitosan lamellae in MOF-aerogel-2. All aerogels, with or without immobilized MOF nanoparticles, were capable of removing carotenoids during oil bleaching. MOF-aerogel-2 showed the most satisfying removal proportions of 26.6%, 36.5%, and 47.2% at 50 °C, 75 °C, and 100 °C, respectively, and its performance was very similar to that of commercial activated clay. The reuse performance of MOF-aerogel-2 was tested, and the results showed its exceptional sustainability for carotenoid removal. These findings suggested the effectiveness of the MOFaerogel for potential utilization in oil bleaching treatments.

## 1. Introduction

Crude vegetable oils are usually unacceptable for consumption because of their dark color, undesirable flavor, and low oxidative stability. Oil refining is widely employed to eliminate impurities, hydrocarbons, gums, free fatty acids (FFAs), natural pigments, unpleasant odors, and waxy substances. Oil refining generally includes successive stages of impurity removal, degumming, deacidification, bleaching, deodorizing, and dewaxing [1]. Oil bleaching is imperative for removing excess natural pigments from crude vegetable oils since they are responsible for the undesirable brownish color and oxidative deterioration, both of which enormously retard the utility of oil in the food, cosmetic, and pharmaceutical industries [2].

The dominant natural pigments determined in plant oils are carotenoids and chlorophylls [3]. Adsorptive bleaching by bleaching earth is the most efficient method with the lowest operational cost through physical interfacial adsorption of these natural pigments [4]. In most cases, the undesired pigments are physically entrapped inside the pores and adhered to the surface of bleaching earth powders by van der Waals forces [4,5]. Common bleaching earth powders, such as diatomite, bentonite, acidic clay, and attapulgite, have been widely investigated as a means of removing natural pigments from vegetable oils [6].

However, after oil bleaching, the presence of oil residual and solvent-rich components greatly increases the potential risk of spontaneous combustion of the spent bleaching earth powders, which is a fire hazard [7]. Proceeding from an inexpensive aspect, most of the spent bleaching earth powders are disposed of in landfills without further treatment or oil recovery, causing contamination of the soil and groundwater through environmental degradation and damage to living organisms [1,7,8]. In addition, the production of disposable bleaching earth usually requires chemical modifications, which may generate massive contaminants during production [6]. Therefore, the development of a novel alternative bleaching agent with high bleaching capability and eco-friendly properties has attracted extensive attention.

In the past two decades, the design and fabrication of functional metal organic frameworks (MOFs) with microporous structures have been intensively investigated for the adsorption of guest molecules [9]. In light of their extremely large surface area, multifunctional ligands, and excellent reusability, the synthesis of MOFs has generated great enthusiasm based on their capacity to entrap and adsorb molecular compounds in different liquid phases. The utilization of MOFs in the lipophilic phase has attracted attention due to their superb performance in essential oil delivery [9], deacidification of crude vegetable oil [10], adsorptive desulfurization in fuel [11], removal of thiophene from n-octane [12], and oil/water separation [13]. It is worth noting that there is a lack of scientific details about lipophilic pigment adsorption onto MOFs during oil bleaching. However, MOFs are mostly microscopic and colloidal crystals, which result in low efficiency for practical applications due to their aggregation tendency, problematic separation, and tedious recycling from the liquid phase. Therefore, it is crucial to load or immobilize MOF nanoparticles on macroscopic polymeric materials or substrates to address these intractable problems.

Among the various MOFs, the UiO-66 series has been extensively investigated for its excellent chemical, aqueous, and thermal stability [14]. This Zr-MOF could be further pre-functionalized through the substitution of the benzene hydrogen of the H_2_BDC organic linker and derived into carboxylic UiO-66. In the current study, carboxylic zirconium MOF (UiO-66-COOH) was chosen for oil bleaching due to its low toxicity [15], facile synthesis [16], and satisfactory stability under thermal conditions. To address the recycling problems of the powder UiO-66-COOH, an eco-friendly chitosan matrix with rich hydroxyl and amino structures was introduced into the gel-state UiO-66-COOH prior to its dehydration, and homogeneous hydrogels were further blended by electrostatic interaction and hydrogen bonds. The complexed hydrogels were converted to MOF-aerogels following lyophilization. We tested the structural and physical-chemical characteristics of the MOF-aerogels and subsequently investigated their soybean oil bleaching performance, reusability, and regenerative properties. This work revealed the changes in the structural effects of the aerogel matrix after the introduction of the UiO-66-COOH and their capability to enrich the oil bleaching performance, further providing a new method for oil refining.

## 2. Experimental

### 2.1. Materials

Chitosan (middle viscosity; 200–400 mPa·s), zirconium tetrachloride (ZrCl_4_; 98% purity), 1,2,4-benzenetricarboxylic acid (H_3_BTC; 98% purity), glacial acetic acid (CH_3_COOH; 99.5% purity), and *n*-hexane (C_6_H_14_; 97%) were obtained from Macklin Biochemical Ltd. (Shanghai, China). All other chemicals and reagents used in this study were of analytical grade. Neutralized soybean oil was obtained from a local processor and stored at 4 °C to avoid oxidation. Commercial bleaching earth, activated clay, was purchased from Haixing Ltd. (Zhengzhou, China).

### 2.2. Preparation of MOF-Aerogels

UiO-66-COOH was synthesized with a previously reported environmentally benign hydrothermal reflux strategy [16]. ZrCl_4_ (4.8 g, 20 mmol) and H_3_BTC (4.4 g, 20 mmol) in 100 mL deionized water were dispersed in a round-bottomed flask by means of stirring (500 rpm). The mixture was heated and refluxed at 100 °C for 24 h. The obtained crude carboxylic Zr-MOF gel was then dissolved and washed using deionized water, followed by an additional 16 h reflux. The uniform colloidal UiO-66-COOH was harvested by centrifugation and then dispersed as a 20 mL MOF suspension with deionized water to prepare the MOF-aerogels.

For the preparation of the MOF-aerogels, chitosan solution (2%, *w*/*v*) was prepared by dissolving chitosan in aqueous acetic acid (1%, *v*/*v*). Then, 150 mL of chitosan solutions were blended with 5 mL and 10 mL of the as-prepared colloidal UiO-66-COOH under ultrasonic treatment (40 kHz). The two mixtures were further freeze-thawed three times and then lyophilized, resulting in MOF-aerogel-1 and MOF-aerogel-2 with different UiO-66-COOH contents. Chitosan aerogel (CS-aerogel) was obtained as a control through lyophilization of the as-prepared chitosan solution.

### 2.3. Characterizations

#### 2.3.1. Wide-Angle X-ray Diffraction

Wide-angle X-ray diffraction (XRD) measurements of the UiO-66-COOH powder, pristine chitosan aerogel, and two MOF-aerogels were firstly pressed into tablets and then conducted with an Xpert3 X-ray diffractometer (Malvern Panalytical, Malvern, UK) utilizing CuKα radiation in a scan range of 2θ from 5–55°.

#### 2.3.2. Fourier Transform Infrared Spectroscopy

Fourier transform infrared (FTIR) spectra of the aerogel and MOF samples in the 4000–400 cm^−1^ wavenumber range were recorded with a Vertex70 spectrometer (Bruker Daltonik, Bremen, Germany).

#### 2.3.3. X-ray Photoelectron Spectroscopy

The mechanism of MOF-aerogel formation was characterized to obtain the high-resolution spectra of O1s using X-ray photoelectron spectroscopy (XPS) with a Kratos Axis Ultra DLD system (Shimadzu, Kyoto, Japan) using AlKα radiation. 

#### 2.3.4. Morphological Studies

Morphological differences between the chitosan aerogel and MOF-aerogel-2 were observed using a Zeiss EVO18 scanning electron microscopy (SEM) system (Carl Zeiss AG, Oberkochen, Germany). Energy dispersive spectroscopy (EDS) as well as elemental mapping of the two MOF-aerogels was performed using an X-Max energy dispersive X-ray system (Oxford Instruments, Abingdon, UK), which was connected with the Zeiss EVO18 SEM system.

#### 2.3.5. Thermal Properties

Thermal gravimetric analysis (Netzsch GmbH, Selb, Germany) was used to characterize the thermal properties as well as stability of the samples at a heating rate of 15 °C/min in a testing temperature range of 30–800 °C under a nitrogen atmosphere. 

#### 2.3.6. X-ray Microtomography Reconstructions

3D reconstructed structures of MOF-aerogel-1 and MOF-aerogel-2 were obtained from X-ray microtomography (micro-CT) analysis, which was carried out in a Metris X-Tek XTV160H instrument operating at 75 kV for detecting the zirconium element within the MOF-aerogels. The cylindrical aerogel samples were scanned for a three-dimensional area of 3 mm × 3 mm × 3 mm.

### 2.4. Oil Bleaching Studies

Oil bleaching experiments were performed at various temperatures to figure out the removal behavior of MOF-aerogels. Typically, the aerogels (0.2 g) and activated clay were added to neutralized soybean oils (20 mL), and the bleaching process was maintained for 1 h at temperatures of 50 °C, 75 °C, and 100 °C in an oil bath. The stirring rate was 300 rpm via a Hei-PLATE stirring system (Heidolph Instruments GmbH, Walpersdorfer, Germany). After the bleaching process, the oil samples were diluted with 4 volumes of *n*-hexane, and their full spectrum was obtained by UV-1900 UV-vis spectrometry (Shimadzu, Kyoto, Japan). The absorbance of the carotenoids in the samples was measured 3 times at their maximum-intensity wavelength of 446 nm; the removal performance of the carotenoids can be calculated with the following equation:(1)$$Removal\;\%=[(A_{0}-A_{t})/A_{0}]\times 100\%$$
where A_0_ represents the absorbance of the raw neutralized soybean oil and A_t_ indicates the absorbance of the bleached oil. The MOF-aerogels were washed using n-hexane and chloroform to remove the residual oil and carotenoid molecules and perform 5 continuous bleaching-desorption cycles to figure out their reusability.

## 3. Results and Discussion

### 3.1. Structural Study

The crystalline structures of the synthesized UiO-66-COOH, CS-aerogel, MOF-aerogel-1, and MOF-aerogel-2 were characterized by XRD (Figure 1). For all aerogel networks, the characteristic peak at 11.59° was attributed to the presence of a hydrated chitosan crystalline skeleton [17,18]. All of the MOF-containing aerogels (MOF-aerogel-1 and MOF-aerogel-2) presented the specific diffraction patterns of the UiO-66 series MOFs in the range of 7.2~8.4° [19,20], in which the single broad peaks observed in the patterns of the two MOF-aerogels were attributed to the presence of the amorphous chitosan substrate. The broad MOF characteristic peaks also indicate the interaction of MOFs with amorphous chitosan chains. This diffraction pattern suggested that the MOF-aerogels sustained the intrinsic topological structures of both the chitosan aerogel and UiO-66-COOH; hence, this result indicated that the MOF was successfully anchored to the chitosan aerogel.

The microscopic structures of the UiO-66-COOH, CS-aerogel, MOF-aerogel-1, and MOF-aerogel-2 were determined in order to understand the interactions between MOF and CS based on their FTIR spectra, and the results are shown in Figure 2. The stretching vibration of UiO-66-COOH at 1400 cm^−1^ was assigned to the conjugated C=C within the benzene ring of the organic linker H_3_BTC, and the adsorptive peak at 1589 cm^−1^ was ascribed to the C-O signal in the carboxyl group [21,22]. The characteristic band at 1710 cm^−1^ confirmed the existence of free carboxyl groups on the external surface of the UiO-66-COOH layer [23]. Among the aerogel samples, the spectra of the CS-aerogel and MOF-aerogel were almost identical in most peak positions because the large-scale chitosan polymeric skeleton plays a role as the main framework of aerogels. Compared to the CS-aerogel, the MOF-aerogels displayed a weak signal at 1370~1366 cm^−1^, which was attributed to the presence of the conjugated π-bonds in the benzene ring [11]. Moreover, the peaks observed in the MOF-containing samples at approximately 654~658 cm^−1^ were ascribed to the Zr-O bond [24]. These observations may possibly indicate the affinity between MOF- and CS-aerogel substrates.

The elemental valence states of the aerogel samples were investigated to test the formation mechanism of MOF-aerogel using XPS spectra. As shown in Figure 3A, the O1s spectrum of the CS-aerogel contained three peaks at 530.16, 531.2, and 533.2 eV, which were respectively associated with NH (C=O) [25], C-OH, and O-C=O/C-O-C [26]. After introducing the gel-state UiO-66-COOH, the O1s spectra of the two MOF-aerogels shifted to lower binding energies (Figure 3B). This shift was due to the introduction of Zr-O bonds and the presence of strong interactions between zirconium and oxygen-containing ligands [27]. Moreover, the strong electrostatic interaction between the glucosamine NH_3_^+^ and the free carboxyl groups in UiO-66-COOH may also change the peak position.

The surface morphological analysis of the CS-aerogel and MOF-aerogel-2 was conducted by SEM, and the images are displayed in Figure 4A–C. The resulting aerogels with or without MOF particles all showed multi-chamber-like lamellar structures with considerable wave-like chitosan flakes stacked into a layered conformation (Figure 4A,B). The two aerogels also exhibited porous structures with various pore sizes, which might be induced by the irregular growth of ice crystals during the freeze-thaw procedure and lyophilization [13]. The magnified view of the MOF-aerogel-2 in Figure 4C shows the presence of numerous polyhedral nanoparticles. After being covered by the carboxylic Zr-MOF nanocrystals, the external surface of the CS-aerogel became rougher, further illustrating that the MOF-aerogel composite was successfully obtained. Elemental distribution analysis of the MOF-aerogel composites conducted by energy dispersive spectroscopy elemental mapping showed that elements of carbon, nitrogen, oxygen, and zirconium were homogeneously distributed throughout the whole chitosan aerogel surface (Figure 4D,E) [28]. The observation of nitrogen and zirconium in the EDS map was due to the existence of chitosan and UiO-66-COOH, respectively. Figure 5 shows a scheme of multiscale structure in the ultra-light MOF-aerogel where the micron-sized carboxylic Zr-MOF were adhered to the surface of CS layers through electrostatic interaction. The interior surface of CS-aerogel tunnels contained a large amount of the glucosamine NH_3_^+^ group, which facilitated the interaction with carboxylic Zr-MOF under electrostatic absorptions. Consequently, the nanosized UiO-66-COOH particles adhered to the internal and external surfaces of the CS-aerogel skeleton.

Thermal gravimetric analysis (TGA) was utilized to determine the thermal properties and stabilities of the UiO-66-COOH, CS-aerogel, MOF-aerogel-1, and MOF-aerogel-2 samples, and the results are shown in Figure 6. The CS-aerogel showed the lowest residue content of 26.4%. The residue contents of MOF-aerogel-1 and MOF-aerogel-2 significantly increased to 28.8% and 31.8%, respectively, indicating that the MOF-aerogels were more difficult to decompose than the CS-aerogel [17]. In addition, the residue content of the pristine MOF was 35.0%. The high residue contents of MOF-aerogels demonstrated that the CS-aerogel showed great potential in immobilizing MOF nanomaterials. In addition, the introduction of MOFs could potentially strengthen the thermal stability of the CS-aerogel.

The three-dimensional morphological characterization of the MOF-aerogels with different UiO-66-COOH contents was performed by X-ray microtomography (micro-CT). As shown in Figure 7, the computed tomography of all samples revealed a sponge-like morphology with interconnected pores throughout the aerogels. The higher X-ray attenuation regions (red color), which are correlated with higher electronic density, were assigned to the UiO-66-COOH nanoparticles. Chitosan-rich regions with a lower X-ray adsorption ability appeared with gray colors in the 3D reconstructed image. The lowest X-ray attenuation regions (black color) of the matrix were assigned to the air trapped within the interconnected porous structures. In the MOF-aerogel-1 sample, an 18.5% bulk volume of the UiO-66-COOH region was observed, while the chitosan and air regions occupied 48.6% and 34.4% of the entire aerogel matrix, respectively (Figure 7A1). In MOF-aerogel-2, approximately 54.5% of the aerogel was coated with the MOF, and the chitosan region without MOF was 34.3% (Figure 7B1). The proportion of air within MOF-aerogel-1 was much higher than that within MOF-aerogel-2. The results of the initial weight loss of water measured from the TGA thermogram (Figure 6) demonstrated the greater affinity of MOF-aerogel-1 to moisture. The strong interactions between glucosamine NH_3_^+^ and free carboxyl groups during the synthesis of MOF-aerogel-2 may decrease the amount of hydrogen bonds, further influencing the moisture content in the composite. The two-dimensional slice images of the right, front, and transverse sections in Figure 7A2–A4 show the densely aggregated structure of the UiO-66-COOH region within MOF-aerogel-1. Slight aggregation of the MOF was observed in the counterpart sections of MOF-aerogel-2 (Figure 7B2–B4). This result indicated that the Zr-MOF particles were coated uniformly on the surface of the CS-aerogel skeleton.

### 3.2. Oil Bleaching under Laboratory Conditions

Oil bleaching is usually performed after the elimination of impurities, hydrocarbons, gums, and free fatty acids. In this paper, we chose neutralized soybean oil to perform bleaching treatments, and UV-vis spectrometric analysis was conducted to test the bleaching capacity of the aerogel samples and figure out the removal behaviors, the results of which are displayed in Figure 8. Deacidified, unrefined soybean oil has a maximum absorbance peak at 446 nm, attributed to the presence of carotenoids [29]. The weak peak at 670 nm indicates minor amounts of chlorophyll [30]. Raw soybean oil samples were treated at 50 °C, 75 °C, and 100 °C with the different aerogels, including pristine CS-aerogel, MOF-aerogel-1, and MOF-aerogel-2. Activated clay, a popular commercial bleaching earth, was also subjected to different treatments.

The majority of natural pigments in soybean oils are carotenoid and chlorophyll, and because the content of the latter was rather low, the adsorption capacity of carotenoid was utilized to evaluate the bleaching processes of soybean oil. Similar spectrophotometric methods were previously reported and performed to test the bleaching capacity of rapeseed oil [29], palm oil [2], and olive oil [31]. Intensities of the peaks of carotenoid at the position of 446 nm were ranked according to the following order: untreated raw oil > 50 °C-treated oil > 75 °C-treated oil > 100 °C-treated oil (Figure 8B–D). This result was due to the increase in temperature, which resulted in the partial degradation of the carotenoids [29]. Notably, thermal treatment at 100 °C showed a minor effect on the degradation of the carotenoids compared to the 75 °C treatment (Figure 8C,D). The adsorbents used for bleaching were ranked according to the carotenoid absorbances under gradient temperature as follows: CS-aerogel treated oil > MOF-aerogel-1 treated oil > MOF-aerogel-2 treated oil > activated clay treated oil. A lower carotenoid absorbance indicates a higher bleaching capacity. Furthermore, a small amount of chlorophyll was still present in the bleached soybean oil. In addition, with increasing temperatures, all bleaching agents showed a higher carotenoid removal ability. These results are also in accordance with the removal data, which are calculated and presented in Figure 9. 

As shown in Figure 9, activated clay showed the best capacity for oil bleaching among all agents. The removal performances of this bleaching earth at 50 °C, 75 °C, and 100 °C were 29.9%, 38.4%, and 50.8%, respectively. Among the aerogels, all samples were capable of removing carotenoids. The MOF-aerogels performed better than the CS-aerogels. This result implied that UiO-66-COOH played a pivotal role in removing natural pigments from soybean oil. It has been widely reported that the bleaching power increases by introducing more surface area, porosity, reactive sites, and surface acidity of bleaching agents [32]. The oil bleaching process, which utilizes bleaching earth with high surface acidity, is generally considered a combination of catalytic and adsorptive reactions to remove natural pigments and undesirable components [33]. The introduction of UiO-66-COOH not only resulted in the introduction of carboxyl groups on the external surface of the chitosan substrate but also enhanced the porosity of the composite aerogel matrix. The removal performances of MOF-aerogel-1 under the 50 °C, 75 °C, and 100 °C treatments were 17.9%, 29.7%, and 45.0%, respectively. The higher MOF loading amount in MOF-aerogel-2 led to satisfying removal proportions of 26.6%, 36.5%, and 47.2% under different temperature gradients, and its performance was very similar to that of commercial activated clay and previously reported acid-activated bentonite [34]. In addition to the MOF content, the arrangement of particles also contributes to the adsorption. The 2D microtomographic images in Figure 7 showed that the MOF particles were distributed homogeneously on the surface of the chitosan lamellae of MOF-aerogel-2, while they aggregated densely within the MOF-aerogel-1 matrix. These different distribution patterns may lead to different macroscopic structural properties and bleaching capacities of the two MOF-aerogels.

The dynamic adsorption processes of bleaching earth and aerogel were different. The mobility of activated clay was much better than that of the aerogels. During the oil bleaching process, the soybean oil was vigorously stirred, and the activated clay particles were homogeneously dispersed with an oil vortex. After the bleaching process, the sludge was separated by filtration or centrifugation and then disposed of in landfills without further treatment, which is usually energy-consuming and causes contamination of the soil and groundwater. The aerogel samples were relatively “inactive”. They were immersed and slowly suspended in soybean oil, and they showed acceptable adsorptive performance. The aerogels used can be easily collected with low energy consumption and without using filters or machines. Owing to the sustainable nature of MOFs, this new hybrid aerogel adsorbent uses a new approach for oil bleaching and can potentially be utilized as a filter fastened to the bleaching reactor.

### 3.3. Reusability of MOF-Aerogels

To meet the sustainable perspective, we further investigated the reuse performance of the MOF-aerogel-2 composite after oil bleaching. Continuous adsorption-desorption cycles were conducted at 100 °C using *n*-hexane and chloroform to remove the residual oil and carotenoid molecules from the composite particles after each test. As shown in Figure 10, MOF-aerogel-2 showed sustainable bleaching capacity (Figure 10A) and retained more than 40% of the removal proportion in the first two reuse cycles (Figure 10B). The reusability of the aerogel decreased to 39.9% in the fourth cycle, slightly lower than the performance of the unused CS-aerogel. Figure 11a shows the raw MOF-aerogel-2, and its counterpart after four reuse cycles is shown in Figure 11b. The MOF-aerogel-2 was ultralight and white. After five adsorption-desorption cycles, MOF-aerogel-2 was slightly yellow and maintained good mechanical strength, and the color could not be removed easily by organic eluents. This indicated that the carotenoid molecules were tightly bound to the binding sites within the MOF particle surfaces. These results suggested the mechanical strength of the porous structure and its remarkable sustainability for carotenoid bleaching. These outcomes demonstrated the efficacy of the carboxylic MOF-aerogel containing hierarchical porous structures for possible application in oil bleaching. In addition, the integration of MOFs and CS substrates endows not only sustainability to micro-sized MOF particles but also expands the application of chitosan.

## 4. Conclusions

Herein, a chitosan-based aerogel was combined with UiO-66-COOH nanoparticles via an easy-handling procedure, leading to the formation of novel MOF-aerogels for efficient oil bleaching. The chitosan aerogel matrix not only provided a porous scaffold to immobilize UiO-66-COOH particles but also increased their affinity through electrostatic interaction with the carboxylic Zr-MOF nanoparticles. The increase in the MOF content not only enhanced the thermal stability of the aerogel matrix but also decreased the agglomeration trend of the UiO-66-COOH nanoparticles, further contributing to a homogeneously blended aerogel structure. The MOF-aerogels maintained the chemical accessibility and structural properties of the carboxylic Zr-MOF nanoparticles, further promoting the oil bleaching capacity. The performance of MOF-aerogel-2 was similar to that of commercial bleaching earth, indicating its potential utilization as a novel aerogel filter. Moreover, the ultralight MOF-aerogel-2 is easy to handle and possesses satisfactory reusability. This new MOF-based material offers an efficient and simple strategy for oil bleaching and may potentially be utilized in lipophilic substance treatment.

## Figures and Tables

**Figure 1 foods-12-04151-f001:**
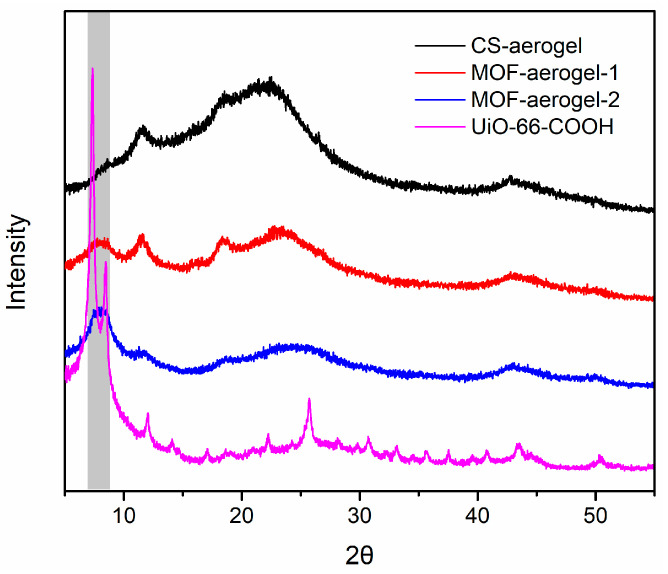
XRD patterns of CS-aerogel, MOF-aerogel-1, MOF-aerogel-2, and UiO-66-COOH nanoparticle.

**Figure 2 foods-12-04151-f002:**
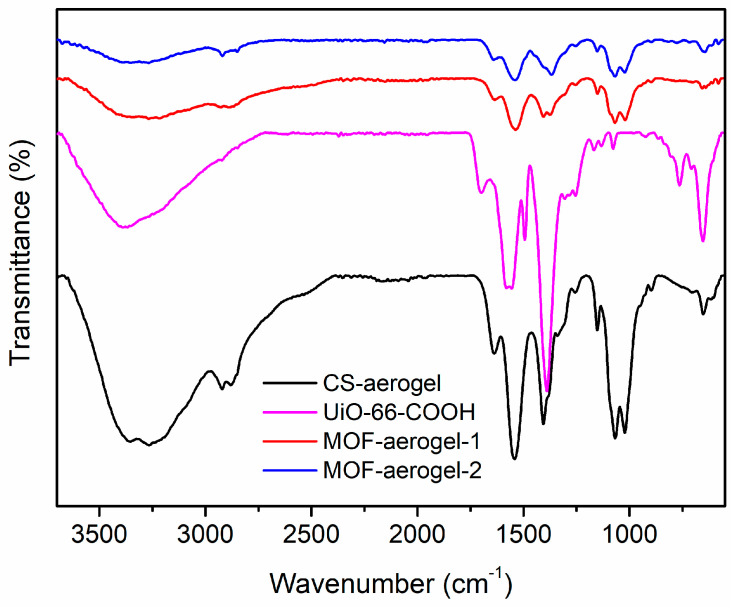
FTIR spectra of CS-aerogel, MOF-aerogel-1, MOF-aerogel-2, and UiO-66-COOH nanoparticles.

**Figure 3 foods-12-04151-f003:**
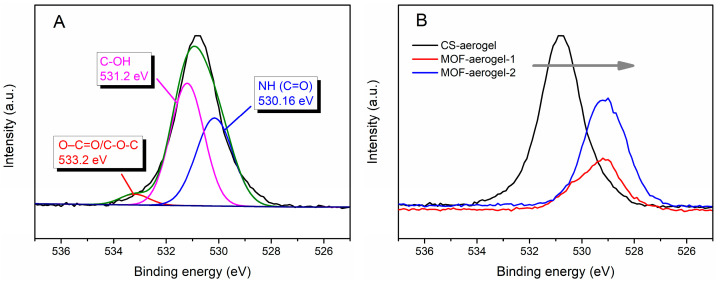
(**A**) High-resolution O1s XPS spectra of CS-aerogel. (**B**) The XPS spectra of O1s for CS-aerogel, MOF-aerogel-1, and MOF-aerogel-2.

**Figure 4 foods-12-04151-f004:**
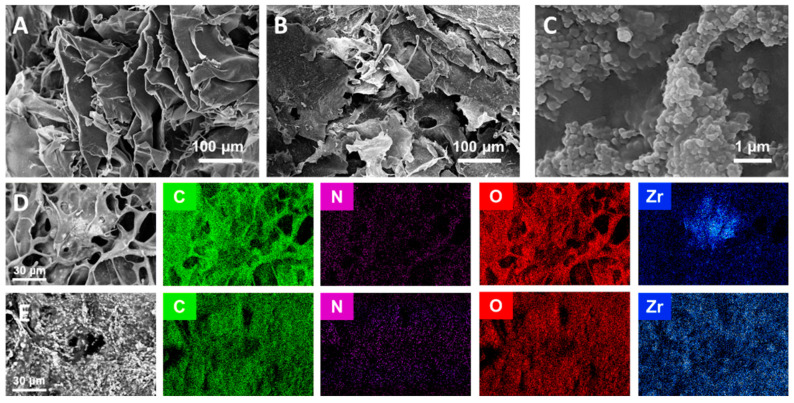
SEM images of (**A**) CS-aerogel, (**B**) MOF-aerogel-2, and (**C**) a high-resolution micrograph of the MOF-aerogel-2 surface. EDX elemental map of (**D**) MOF-aerogel-1 and (**E**) MOF-aerogel-2.

**Figure 5 foods-12-04151-f005:**
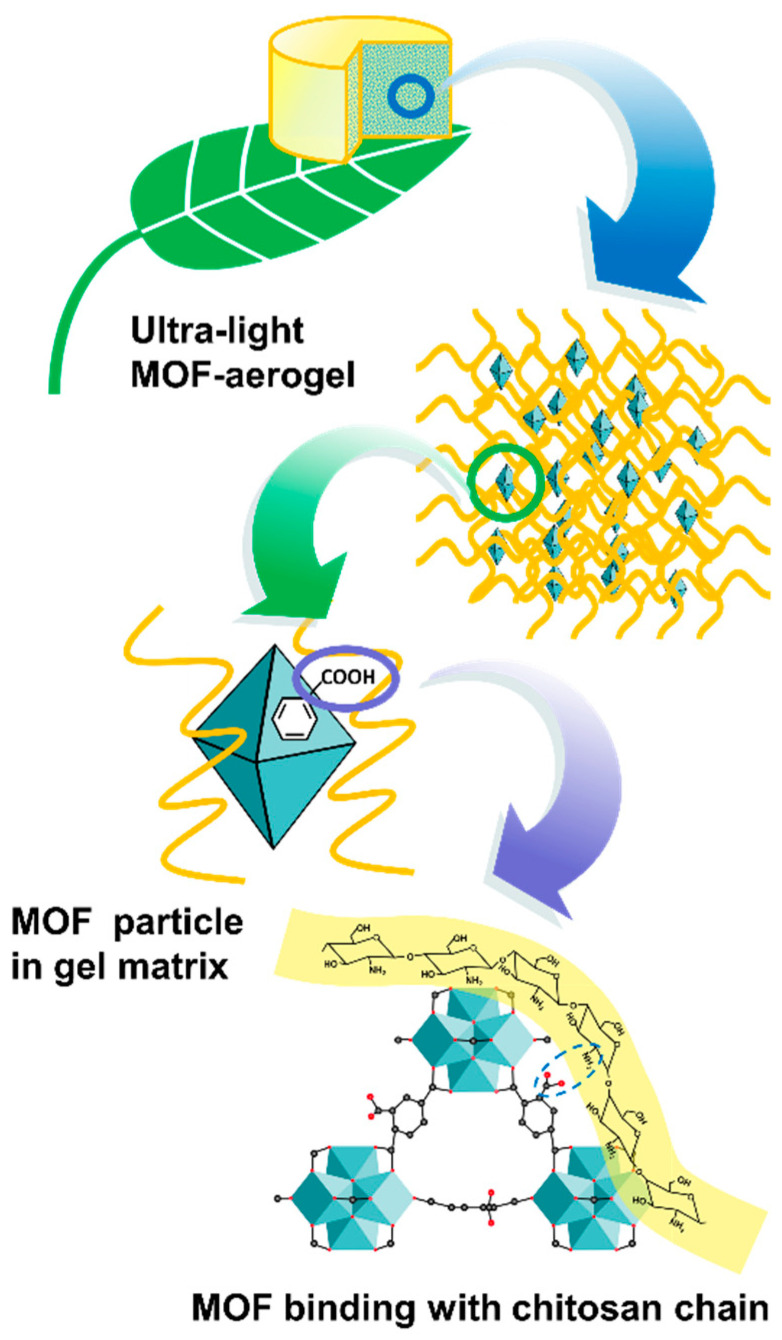
Schematic diagram of the multiscale structure in the MOF-aerogel.

**Figure 6 foods-12-04151-f006:**
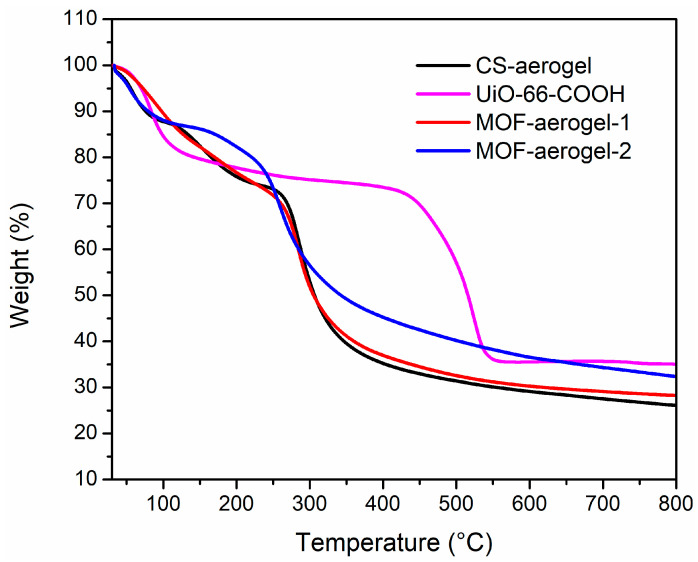
TG curves of CS-aerogel, MOF-aerogel-1, MOF-aerogel-2, and UiO-66-COOH nanoparticles.

**Figure 7 foods-12-04151-f007:**
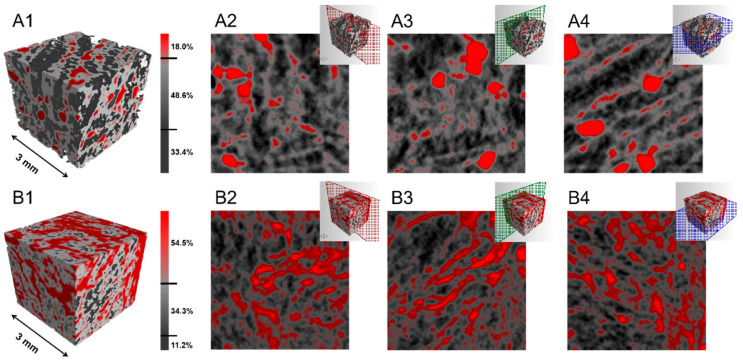
X-ray microtomography reconstructions of (**A1**) MOF-aerogel-1 and (**B1**) MOF-aerogel-2; two-dimensional side slice images in right, front, and transverse sections of (**A2**–**A4**) MOF-aerogel-1 and (**B2**–**B4**) MOF-aerogel-2.

**Figure 8 foods-12-04151-f008:**
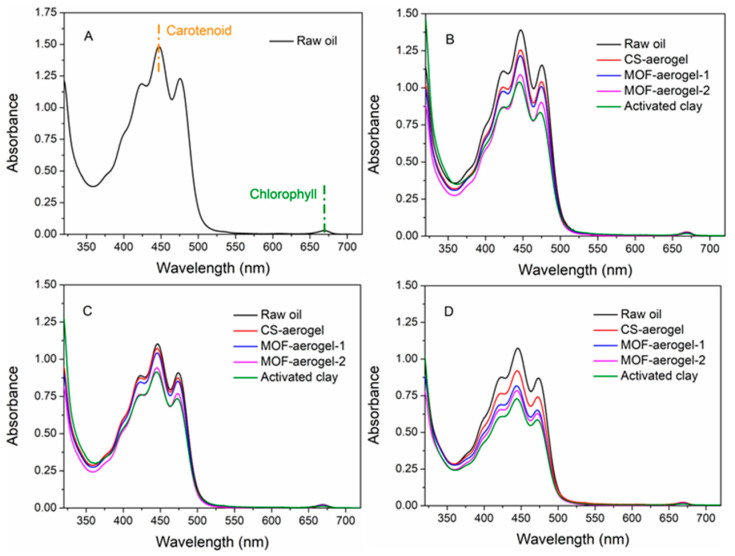
UV-vis spectra of (**A**) raw neutralized soybean oil and neutralized soybean oil treated with different bleaching agents under (**B**) 50 °C, (**C**) 75 °C, and (**D**) 100 °C.

**Figure 9 foods-12-04151-f009:**
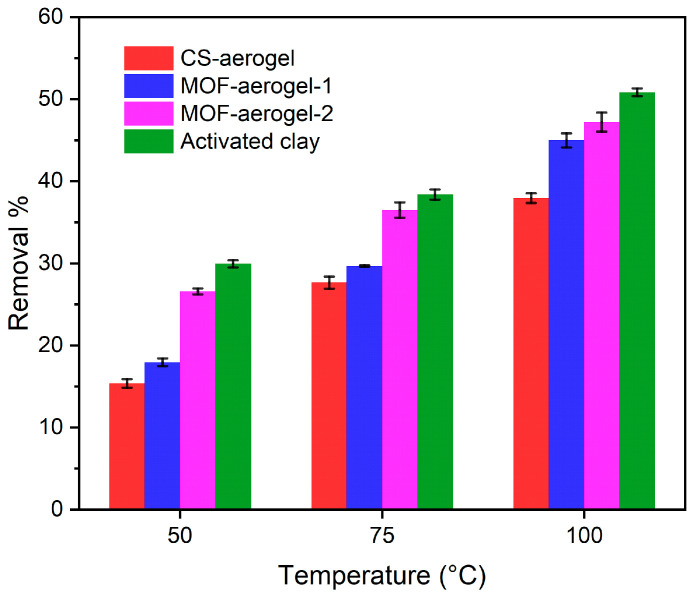
Carotenoid removal performance from soybean oil of different bleaching agents treated under 50 °C, 75 °C, and 100 °C.

**Figure 10 foods-12-04151-f010:**
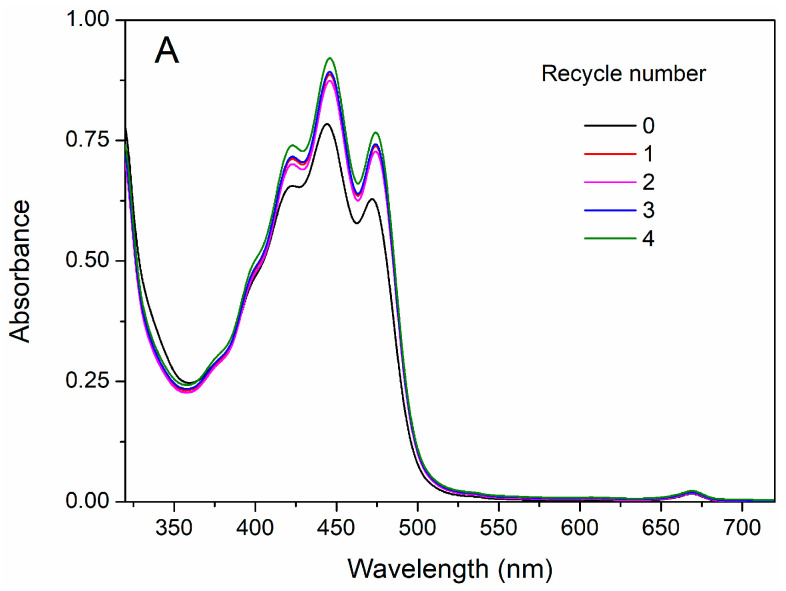
UV-vis spectra of bleached soybean oil (**A**) and carotenoid removal proportion (**B**) after multiple reuse cycles of MOF-aerogel.

**Figure 11 foods-12-04151-f011:**
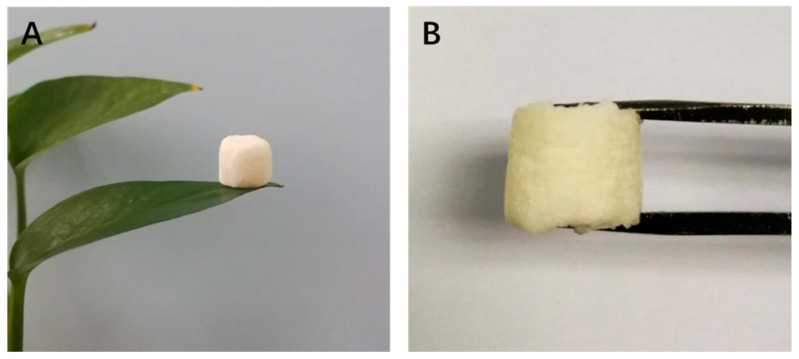
(**A**) Photograph of ultra-light MOF-aerogel-2. (**B**) Photograph of MOF-aerogel-2 after five adsorption-desorption cycles.

## Data Availability

The data presented in this study are available on request from the corresponding author.

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
