# Peer review of "Design of Recyclable Carboxylic Metal-Organic Framework/Chitosan Aerogels for Oil Bleaching"

_foods, 2023, doi:10.3390/foods12224151_

Round 1
Reviewer 1 Report
Comments and Suggestions for Authors
An interesting work is presented primarily from a synthetic point of view. For a more complete understanding of the supramolecular structure of MOF-СS, it would be useful to study the surface structure not only by SEM, but also by AFM. This method allows one to obtain more detailed information about the features of complex hierarchical supramolecular structures.
In this paper, new hierarchical organometallic skeleton/chitosan aerogel compositions were developed for the first time for oil bleaching. UiO-66-COOH-type carboxylic zirconium metal organic frame- works (Zr-MOFs) were synthesized and immobilized onto a chitosan matrix with different contents and named MOF-aerogel-1 and MOF-aerogel-2.
Among the various MOFs, the UiO-66 series has been extensively investigated for their excellent chemical, aqueous and thermal stability. In the current study, carboxyl- functionalized UiO-66 (UiO-66-COOH) was chosen for oil bleaching due to its nontoxicity, facile synthesis, and satisfactory stability under thermal conditions. To address the recycling problems of the powder UiO-66-COOH, a chitosan hydrogel matrix was in- troduced into the gel-state UiO-66-COOH prior to its dehydration, and homogeneous hydrogels were further blended by electrostatic interaction. The complexed hydrogels were converted to MOF aerogels following lyophilization. In this paper, the structural and physical properties of the MOF aerogels were tested, and subsequently investigated their soybean oil bleaching performance, reusability and regenerative properties.
It has been shown, the chitosan aerogel matrix not only provided a porous skeleton to accommodate UiO-66-COOH particles but also increased their affinity through electrostatic interaction with the carboxylic Zr-MOF nanoparticles. The MOF aerogels preserved the intrinsic accessibility and structural properties of the UiO-66-COOH nanoparticles. The increase in the MOF content not only enhanced the thermal stability of the aerogel matrix but also decreased the aggregation tendency of the MOF nanoparticles, further contributing to a homogeneously blended aerogel structure. This structure preserved the accessibility of UiO-66-COOH within the aerogel composite, further promoting the oil bleaching capacity. The performance of MOF-aerogel-2 was similar to that of commercial bleaching earth, indicating its potential utilization as a novel aerogel filter. Moreover, the ultralight MOF-aerogel-2 is easy to handle and possesses satisfactory reusability. In particular, continuous adsorption-desorption cycles were conducted at 100°C using n-hexane and chloroform to remove the residual oil and carotenoid molecules from the composite particles after each test. It was shown, the MOF-aerogel-2 showed sustainable bleaching capacity and retained more than 40% of the removal rate in the first two reuse cycles. The reusability of the aerogel decreased to 39.9% in the fourth cycle, slightly lower than the performance of the unused CS-aerogel.
The presented results convincingly show that the new MOF-based material may potentially be utilized in lipophilic substance treatment.
Author Response
Reviewer 1: Comments and Suggestions for Authors
An interesting work is presented primarily from a synthetic point of view. For a more complete understanding of the supramolecular structure of MOF-СS, it would be useful to study the surface structure not only by SEM, but also by AFM. This method allows one to obtain more detailed information about the features of complex hierarchical supramolecular structures.
In this paper, new hierarchical organometallic skeleton/chitosan aerogel compositions were developed for the first time for oil bleaching. UiO-66-COOH-type carboxylic zirconium metal organic frame- works (Zr-MOFs) were synthesized and immobilized onto a chitosan matrix with different contents and named MOF-aerogel-1 and MOF-aerogel-2.
Among the various MOFs, the UiO-66 series has been extensively investigated for their excellent chemical, aqueous and thermal stability. In the current study, carboxyl- functionalized UiO-66 (UiO-66-COOH) was chosen for oil bleaching due to its nontoxicity, facile synthesis, and satisfactory stability under thermal conditions. To address the recycling problems of the powder UiO-66-COOH, a chitosan hydrogel matrix was in- troduced into the gel-state UiO-66-COOH prior to its dehydration, and homogeneous hydrogels were further blended by electrostatic interaction. The complexed hydrogels were converted to MOF aerogels following lyophilization. In this paper, the structural and physical properties of the MOF aerogels were tested, and subsequently investigated their soybean oil bleaching performance, reusability and regenerative properties.
It has been shown, the chitosan aerogel matrix not only provided a porous skeleton to accommodate UiO-66-COOH particles but also increased their affinity through electrostatic interaction with the carboxylic Zr-MOF nanoparticles. The MOF aerogels preserved the intrinsic accessibility and structural properties of the UiO-66-COOH nanoparticles. The increase in the MOF content not only enhanced the thermal stability of the aerogel matrix but also decreased the aggregation tendency of the MOF nanoparticles, further contributing to a homogeneously blended aerogel structure. This structure preserved the accessibility of UiO-66-COOH within the aerogel composite, further promoting the oil bleaching capacity. The performance of MOF-aerogel-2 was similar to that of commercial bleaching earth, indicating its potential utilization as a novel aerogel filter. Moreover, the ultralight MOF-aerogel-2 is easy to handle and possesses satisfactory reusability. In particular, continuous adsorption-desorption cycles were conducted at 100°C using n-hexane and chloroform to remove the residual oil and carotenoid molecules from the composite particles after each test. It was shown, the MOF-aerogel-2 showed sustainable bleaching capacity and retained more than 40% of the removal rate in the first two reuse cycles. The reusability of the aerogel decreased to 39.9% in the fourth cycle, slightly lower than the performance of the unused CS-aerogel.
The presented results convincingly show that the new MOF-based material may potentially be utilized in lipophilic substance treatment.
Answer: Thank you for your kind advice. We have revised our manuscript carefully and listed corrections point by point.
Reviewer 2 Report
Comments and Suggestions for Authors
The manuscript discuss the use of MOF/chitosan aerogels for the bleaching of a vegetable oil. Here are my comments that should be addressed before accepting the paper for publication:
A lot of English mistakes should be corrected, for instance: Line 62, 63 “In light of their extremely large surface area, multifunctional ligands, extremely high surface area and excellent reusability,…”, please delete the repetition – Line 134 “Removal rate” change to Removal % – Line 152 “The chemical groups of the samples were analyzed according to their FTIR spectra “ correct, Line 205, 210 and 212 “loading rate“ change to loading% , ..
-Please explain how the XRD patterns of aerogels were measured, were the aerogels ground to powder or measured as it is?
-The XRD interpretation: “All of the MOF-containing aerogels (MOF-aerogel-1 and MOF aerogel-2) showed the typical diffraction patterns of the UiO-66” Please reconsider and add more discussion.
-FTIR analysis: please normalize the intensity of the different spectra, add the values of the peaks on the spectra, and recheck the interpretation.
- How the TGA was studied, in the air or under nitrogen? if in the air why no full decomposition of chitosan was achieved, if under nitrogen, is it possible to calculate the loading content in these conditions?
-Please support your assumption that interaction with UiO-66-COOH via the formation of
electrostatic interactions (Figure 5.) by FTIR discussion.
- “The higher air content was attributed to the higher moisture content in MOF-aerogel-1 before lyophilization” How? please correct.
- “This result indicated that the Zr-MOF nanoparticles were distributed uniformly on the surface of the MOF-aerogel-1 skeleton”. Please correct
-Figure 9: change removal rate to removal % and in the text everywhere.
- Please add the experiments of pristine Zr-MOF nanoparticles for the oil bleaching as a reference for comparison.
- Please recheck/recalculate the removal% presented in Figure 9, the spectra in Figure 8 show less than the stated values.
-“this is the first time that MOF-based materials have been utilized for oil bleaching” Please change this sentence, and show the difference between your work and similar work in literature; for instance : Food Chemistry, Volume 190, 1 January 2016, Pages 103-109 and Structure and properties of selected metal organic frameworks as adsorbent materials for edible oil purification, LA RIVISTA ITALIANA DELLE SOSTANZE GRASSE - VOL. XCVI - GENNAIO / MARZO 2019“.
-Please move the Figures in Supporting Information to the main manuscript.
Comments on the Quality of English Language
English typos and grammar mistakes should be corrected.
Author Response
Reviewer 2: Comments and Suggestions for Authors
The manuscript discusses the use of MOF/chitosan aerogels for the bleaching of a vegetable oil. Here are my comments that should be addressed before accepting the paper for publication:
Answer: Thank you for your kind advice. We have revised our manuscript carefully and listed corrections point by point.
A lot of English mistakes should be corrected, for instance: Line 62, 63 “In light of their extremely large surface area, multifunctional ligands, extremely high surface area and excellent reusability, …”, please delete the repetition
Answer: The mistakes have been revised accordingly.
– Line 134 “Removal rate” change to Removal %
Answer: All terms of the “removal rate” have been changed accordingly.
– Line 152 “The chemical groups of the samples were analyzed according to their FTIR spectra “correct, Line 205, 210 and 212 “loading rate “change to loading%, ..
Answer: It has been revised accordingly.
-Please explain how the XRD patterns of aerogels were measured, were the aerogels ground to powder or measured as it is?
Answer: It has been revised accordingly. Wide angle X-ray diffraction (XRD) measurements of the UiO-66-COOH powder, pristine chitosan-aerogel, and two MOF-aerogels were firstly squashed into tablets and then conducted with an Xpert3 X-ray diffractometer (Malvern Panalytical, Malvern, UK) utilizing CuKα radiation (λ = 0.15406 nm) in a scan range of 2θ from 5–55°.
-The XRD interpretation: “All of the MOF-containing aerogels (MOF-aerogel-1 and MOF aerogel-2) showed the typical diffraction patterns of the UiO-66” Please reconsider and add more discussion.
Answer: It has been revised accordingly.
-FTIR analysis: please normalize the intensity of the different spectra, add the values of the peaks on the spectra, and recheck the interpretation.
Answer: The intensity of peaks in FTIR spectra does not reflect structural information. The extent of redshift and blueshift indicates whether an interaction happens or not. The blue shift can be caused by group substitutions, which shorten the maximum absorption wavelength. Redshift often occurs when molecules are conjugated, co-chromophores or chromophores are introduced.
- How the TGA was studied, in the air or under nitrogen? if in the air why no full decomposition of chitosan was achieved, if under nitrogen, is it possible to calculate the loading content in these conditions?
Answer: We performed TGA experiments under a nitrogen atmosphere. Under this circumstance, it is not accurate to calculate the MOF content. We thus removed the corresponding discussion.
-Please support your assumption that interaction with UiO-66-COOH via the formation of electrostatic interactions (Figure 5.) by FTIR discussion.
Answer: In the XPS study, after introducing the UiO-66-COOH, the O1s spectra of the two MOF-aerogels shifted to lower binding energies (Figure. 3B). This shift was due to the introduction of Zr-O bonds and the electrostatic interaction between the glucosamine NH3+ and the free carboxyl groups in UiO-66-COOH may also change the binding energy. This observation is in accordance with the FTIR result.
- “The higher air content was attributed to the higher moisture content in MOF-aerogel-1 before lyophilization” How? please correct.
Answer: It has been revised accordingly.
Previously, we assumed that MOFs are more hydrophobic than chitosan. We now realize that the MOF colloid also has a large amount of moisture, and potentially increases the water content of the matrix.
- “This result indicated that the Zr-MOF nanoparticles were distributed uniformly on the surface of the MOF-aerogel-1 skeleton”. Please correct
Answer: It has been revised accordingly.
-Figure 9: change removal rate to removal % and in the text everywhere.
Answer: It has been revised accordingly.
- Please add the experiments of pristine Zr-MOF nanoparticles for the oil bleaching as a reference for comparison.
Answer: Actually, we have conducted the adsorption study of UiO-66-COOH before. We thought pristine MOFs could show better performance than their aerogel composites. While we found this nanoparticle, UiO-66-COOH, may not be suitable for direct separation use in oil but suitable for immobilizing onto other substrates. There are some reasons:
(1) Once UiO-66-COOH powder was put into oil, severe aggregation could be observed. While a small part of UiO-66-COOH nano-particles were well dispersed in oil.
(2) After the oil bleaching process, we found some UiO-66-COOH particles were extremely laborious to separate from oil. The presence of nano-fragment changed the maximum absorbing wavelength of matrix (even 10000-rpm’s centrifugation was not sufficient to remove MOF nano-fragments).
(3) Interestingly, when UiO-66-COOH particles were immobilized onto chitosan aerogel, the wavelength at maximum absorbance of solution did not change. This indicates that nano fragments of UiO-66-COOH could be separated with chitosan. Besides, the bleaching capacity towards MOF aerogels was satisfying.
- Please recheck/recalculate the removal% presented in Figure 9, the spectra in Figure 8 show less than the stated values.
Answer: We rechecked the removal performances presented in Figure 9, and the values are in accordance with Figure 8.
-“this is the first time that MOF-based materials have been utilized for oil bleaching” Please change this sentence, and show the difference between your work and similar work in literature; for instance : Food Chemistry, Volume 190, 1 January 2016, Pages 103-109 and Structure and properties of selected metal organic frameworks as adsorbent materials for edible oil purification, LA RIVISTA ITALIANA DELLE SOSTANZE GRASSE - VOL. XCVI - GENNAIO / MARZO 2019“.
Answer: It has been revised accordingly throughout the manuscript.
The paper (Food Chemistry, 2016, 190, 109-109) focused on the deacidification of oil, and this procedure is usually conducted after oil bleaching during the whole oil refining process.
-Please move the Figures in Supporting Information to the main manuscript.
Answer: It has been revised accordingly.
Comments on the Quality of English Language
English typos and grammar mistakes should be corrected.
Answer: Thanks for the referee’s kind advice. English typos and grammar mistakes were revised accordingly.
Reviewer 3 Report
Comments and Suggestions for Authors
The paper describes the synthesis and use of chitosan-based aerogels combined with an MOF, UiO-66-COOH, used for oil bleaching.
The work is experimentally and analytically well structured, I report below some remarks for the authors.
1) line 20-22: this sentence in the abstract is not clear. Please revise.
2) line 61-65: in this section some references to MOFs in the adsorption of guest molecules should be added.
3) In the last part of the introduction the authors introduce chitosan: but why chitosan? The authors should describe and introduce some properties of this polymer.
4) Section 2.3: Characterization - More details about sample preparation and analysis conditions are needed.
5) line 126-129: in the oil bleaching studies, more details on instrumentation are needed: which type of shaker? How do you maintain the temperature at 50 °C, 75 °C and 100 °C? Please complete.
6) line 134: write the equation professionally.
7) line 196: group termini - what does it mean?
8) Equations and Figures should be cited in the text following the journal style.
9) Oil bleaching treatment: what are the common conditions for this treatment? At which temperature is it usually performed?
10) Results and Discussion section: maybe this section could be divided into different paragraphs to help the reader (oil bleaching experiment, reusability tests, and so on...)
11) line 312-321: this section should be better exposed and explained, as the content of this paragraph is an important point technologically speaking.
12) The reuse studies are reported only in the R&D part: please add in the M&M section a detailed description of these tests.
13) Regarding the leaching of the MOF during the recyclability studies: did you conduct any leaching test on the organic solutions used for the regeneration of the material during the different cycles?
Comments on the Quality of English Language
The text of the paper is written quite correctly and fluently, I suggest the authors reread the text to correct just a few minor spells.
Author Response
Reviewer 3: Comments and Suggestions for Authors
The paper describes the synthesis and use of chitosan-based aerogels combined with an MOF, UiO-66-COOH, used for oil bleaching. The work is experimentally and analytically well structured, I report below some remarks for the authors.
Answer: Thank you for your kind advice. We have revised our manuscript carefully and listed corrections point by point.
1) line 20-22: this sentence in the abstract is not clear. Please revise.
Answer: It has been revised accordingly.
2) line 61-65: in this section some references to MOFs in the adsorption of guest molecules should be added.
Answer: It has been revised accordingly.
3) In the last part of the introduction the authors introduce chitosan: but why chitosan? The authors should describe and introduce some properties of this polymer.
Answer: Chitosan is a functional carbohydrate prepared by deacetylation of chitin under alkaline conditions, and the basic unit of its structure is mainly D-glucosamine. Chitosan has a rich hydroxyl and amino structure, which easily reacts with electrophiles to produce derivatives with various functional properties. In recent years, chitosan aerogels have attracted the attention of the research community of biomass porous materials because of their high specific surface area, strong adsorption and good biocompatibility. Chitosan also has a certain inhibitory ability against fungi and bacteria, which is of great significance for the long-term storage of aerogel materials. Previously, we found that chitosan played the role of an affinity agent to strengthen the decoration effect via electrostatic interaction and hydrogen bonding with carboxylic Zr-MOF particles and porous starch, respectively (Carbohydrate Polymers, 2021, 253, 117305). We thus believe chitosan could potentially serve as an ideal substrate for immobilizing UiO-66-COOH.
4) Section 2.3: Characterization - More details about sample preparation and analysis conditions are needed.
Answer: It has been revised accordingly.
5) line 126-129: in the oil bleaching studies, more details on instrumentation are needed: which type of shaker? How do you maintain the temperature at 50 °C, 75 °C and 100 °C? Please complete.
Answer: Corresponding information has been provided accordingly.
6) line 134: write the equation professionally.
Answer: It has been revised accordingly.
7) line 196: group termini - what does it mean?
Answer: It has been revised accordingly.
8) Equations and Figures should be cited in the text following the journal style.
Answer: It has been revised accordingly.
9) Oil bleaching treatment: what are the common conditions for this treatment? At which temperature is it usually performed?
Answer: Oil bleaching uses the absorption effect of absorbents to remove the carotenoid and chlorophyll in the oil. Before oil bleaching, the vegetable oil must be degummed and deacidified to remove moisture and other impurities as much as possible. Oil bleaching usually be carried out at a temperature of 90~125°C and filtered to obtain decolorized oil.
10) Results and Discussion section: maybe this section could be divided into different paragraphs to help the reader (oil bleaching experiment, reusability tests, and so on...)
Answer: It has been revised accordingly.
11) line 312-321: this section should be better exposed and explained, as the content of this paragraph is an important point technologically speaking.
Answer: It has been revised accordingly.
The dynamic adsorption behaviors of MOF-aerogel and commercial clay are different. This study attempts provide a new approach aiming low energy consumption and preventing secondary pollution caused by active clay after oil bleaching.
12) The reuse studies are reported only in the R&D part: please add in the M&M section a detailed description of these tests.
Answer: Corresponding information has been provided accordingly.
13) Regarding the leaching of the MOF during the recyclability studies: did you conduct any leaching test on the organic solutions used for the regeneration of the material during the different cycles?
Answer: Thanks for the referee’s kind advice. In the future, we are going to perform research to determine the metal elements and organic compounds inside the oil and then estimate the leakage of MOFs into oil.
Comments on the Quality of English Language
The text of the paper is written quite correctly and fluently, I suggest the authors reread the text to correct just a few minor spells.
Answer: Thanks for the referee’s kind advice. English typos were revised accordingly.
Round 2
Reviewer 2 Report
Comments and Suggestions for Authors
The Authors have addressed my comments, the manuscript can be accepted in the current form.